# Impacts of Heat and Drought on Gross Primary Productivity in China

**Xiufang Zhu** [1,2,3], **Shizhe Zhang** [1,3,*], **Tingting Liu** [3] and **Ying Liu** [3]

1 State Key Laboratory of Remote Sensing Science, Jointly Sponsored by Beijing Normal University and Institute of Remote Sensing and Digital Earth of Chinese Academy of Sciences, Beijing 100875, China; zhuxiufang@bnu.edu.cn
2 Key Laboratory of Environmental Change and Natural Disaster, Ministry of Education, Beijing Normal University, Beijing 100875, China
3 Institute of Remote Sensing Science and Engineering, Faculty of Geographical Science, Beijing Normal University, Beijing 100875, China; 202021051185@mail.bnu.edu.cn (T.L.); liuying_ly@mail.bnu.edu.cn (Y.L.)
* Correspondence: 202021051191@mail.bnu.edu.cn

**Abstract:** Heat and drought stress, which often occur together, are the main environmental factors limiting the survival and growth of vegetation. Studies on the response of gross primary production (GPP) to extreme climate events such as heat and drought are highly significant for the identification of ecologically vulnerable regions, ecological risk assessments, and ecological environmental protection. We got 1982–2017 climatic data from the University of East Anglia Climatic Research Unit, Norwich, England, and GPP data from National Earth System Science Data Sharing Service Platform, Beijing, China. Using Theil–Sen median trend analysis and the Mann–Kendall test, we analyzed trends in temperature and the standardized precipitation/standardized precipitation evapotranspiration indices in the eight vegetation regions of China. Additionally, the response of GPP to the single and combined impacts of heat and drought were analyzed using multidimensional copula functions, and GPP reduction probabilities were estimated under different drought levels and heat intensities. The results showed that the probability of a drastic GPP reduction increases with increasing drought levels and heat intensities. The combined impacts of heat and drought on vegetation productivity is greater than the impacts of either drought or heat alone and presents a nonlinear superposition of the two extremes. The impact of heat on GPP is not evident when the drought level is high. The temperate grassland and warm temperate deciduous broad-leaved forest regions are the most sensitive regions to drought and heat in China. This study provides a scientific basis for the comprehensive evaluation of the risk of GPP reduction under the single and combined impacts of heat stress and drought stress.

**Keywords:** GPP; heat; SPI; SPEI; copula function

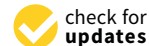

## 1. Introduction

Gross primary production (GPP) refers to the total carbon amount fixed by green vegetation per unit area per unit time through photosynthesis and is the material basis for the survival and development of human society; changes in GPP affect the entire terrestrial carbon cycle [1,2]. $CO_2$ fertilization and extensions of the growing season are expected to enhance vegetation growth because of ongoing global warming [2], and there is evidence of increased vegetation coverage worldwide, even in some semiarid regions [3,4]. However, the frequency, persistence, and magnitude of extreme climate events such as droughts, storms, floods, heat, heat waves, extremely low temperatures, and heavy rains are projected to further increase in the mid-to-late 21st century [1,2,5–7] and may pose potential threats to vegetation growth and terrestrial carbon uptake [8–10]. Understanding the responses of terrestrial GPP to extreme climate events such as heat and drought in the context of the potential aggravation of climatic extremes is of great significance for

predicting the responses of global terrestrial ecosystems to future climatic changes with confidence [11–15].

Drought stress is the main environmental factor limiting the productivity of terrestrial ecosystems [16], and heat stress causes physiological damage to vegetation and reduces vegetation productivity (including crop yield) [17]. Scholars have used GPP, net primary productivity, net ecosystem productivity, and vegetation indices to characterize ecosystem productivity in response to these two main climate anomalies, heat and drought, and have performed a series of related studies on the impacts of drought and heat on terrestrial ecosystem productivity on global and regional scales [1,18–21]. Several global studies have shown that from 2000 to 2010, droughts caused declines in vegetation productivity in most parts of the world [18,22,23]. On a regional scale, frequent drought events in Africa in the past few decades have led to different degrees of famine, causing high life and economic losses and seriously impeding the economic development of some African countries [24–26]. In addition, due to global warming and drought, Australia has recently experienced bushfires of unprecedented severity and scale [27]. Moreover, droughts with different intensities and droughts occurring in different seasons have diverse impacts on different ecosystems (grasslands, forests, farmland, etc.) and vegetation types (broad-leaved forests, coniferous forests, etc.) [19,28–31]. Regarding heat, a study by Ciais et al. [32] showed that insufficient precipitation and an extreme increase in summer heat caused a decrease in vegetation productivity in Eastern and Western Europe in 2003. Wohlfahrt et al. [33] found that GPP decreased linearly under short-term intense heat. Studies focused on grain yield loss under high-temperature conditions have also been performed [34,35].

The sensitivities of vegetation and adverse ecological effects to temperature rising are particularly concerning [36,37], especially as the coincidence of temperature rising with drought intensifies the physiological stress and mortality of global vegetation [38,39]. The impact of the interaction between water availability and temperature on plant physiology is strongly complex. Extremely warm conditions at the beginning of a growing season may compensate for ecosystem carbon losses later during a water deficit; meanwhile, early vegetation activity caused by extremely warm conditions likely also contributes to exacerbating the impacts of drought through reduced initial soil moisture [15]. Moreover, heat and drought often occur at the same time and interact with each other. Heat exacerbates water stress, which in turn increases heat damage. Dong et al. [36] found that the sensitivity of climate warming regions to drought increased significantly in Southern California, indicating that temperature played an important role in increasing vulnerability. Buttlar et al. [15] studied the combination of heat and drought and found that compared with any single-factor extreme, combined heat and drought events led to the strongest observed carbon sink reduction.

At present, researchers often use two kinds of methods to study the impacts of extreme climate events on GPP. One method is to compare and analyze GPP values obtained before and after the occurrence of extreme climate events [32,40,41], and the other method is to first extract the extreme climate events and extreme GPPs and then study the relationship between them through correlation analysis [2,14,15]. Very few studies have utilized copula functions to analyze the impacts of extreme climate events on vegetation GPP. The copula is a probability model that represents a multivariate uniform distribution, and it can be used to examine the association or dependence between many variables [42,43]. It is widely used in statistics, finance, risk management, and other fields [44–46]. Therefore, based on climate (temperature and precipitation) data, GPP data from 1982 to 2017, and China's vegetation zoning vector boundary data, this study analyzed the trends of GPP, temperature, and the drought index (standardized precipitation index (SPI) and standardized precipitation evapotranspiration index (SPEI)) in China over the past 36 years and estimated the probabilities of extreme negative GPP anomalies under different heat and drought conditions by using multidimensional copula functions. This study contributes to the understanding of the possible distribution of GPP under specific climatic conditions, to making appropriate

agricultural production decisions, and to understanding the biochemical dynamics of the global ecosystem carbon cycle.

## 2. Study Region and Data

### 2.1. Study Region

China, 18°10′N–53°33′N, was the study area used in this research with the exclusion of China's South Sea area (Figure 1). There are obvious spatial differences in terrestrial ecosystem GPP in China. High annual GPP values are mainly distributed in the eastern region where there is high vegetation coverage, while low annual GPP values are distributed in the western region where there is low vegetation coverage. Due to differences in vegetation phenology, GPP generally increases from northwest to southeast. The study area was divided into eight subregions according to China's vegetation zoning data. It should be noted that the eight subareas are used for geographical zoning rather than vegetation type zoning. The temperate desert region (R1), the temperate grassland region (R2), and the climatic range of the alpine vegetation region on the Qinghai–Tibet Plateau (R3) are restricted by insufficient precipitation or low temperatures and have sparse vegetation and low GPP levels [47–49]. Precipitation in the temperate coniferous and deciduous forest mixed forest region (R7) and the cold temperate coniferous forest region (R8) is abundant; vegetation growth in these regions is mainly restricted by temperature and radiation, and the vegetation coverage and GPP are relatively high [50]. The warm temperate deciduous broad-leaved forest region (R6) is an agricultural grain-producing area in North China, and the GPP level in this region is higher than that in the surrounding area [51]. The vegetation productivity levels of the subtropical evergreen broad-leaved forest region (R4) and the tropical monsoon forest and rainforest region (R5) are vigorous. Temperature, precipitation, and light are all conducive to vegetation growth, and these factors contribute to the GPP of R5 being the highest among all zones. In addition, GPP in China presents obvious seasonality [51]. The seasonal dynamics of GPP among the different ecological zones present a single-peak type, with the maximum values occurring in summer [52]. The negative impacts of climate change on China's natural ecosystems and agriculture have been widely reported [53–55].

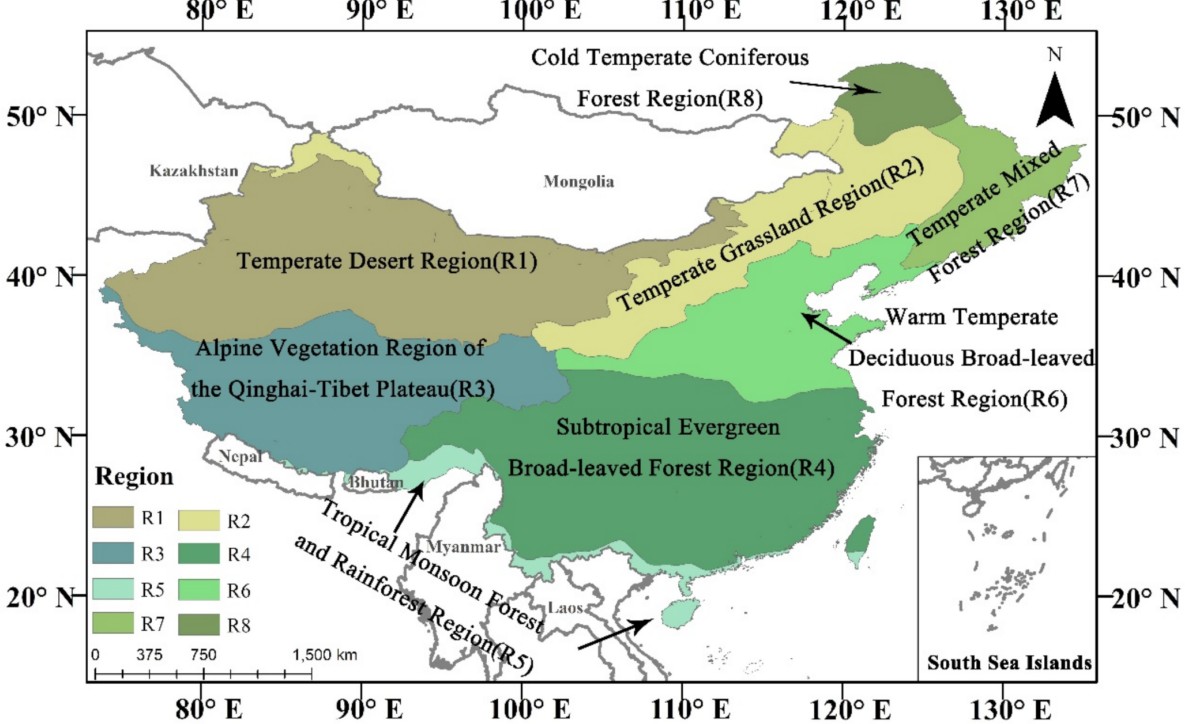

**Figure 1.** Eight vegetation regions in China.

## 2.2. Data

Three main types of data were utilized in this study. First, monthly potential evapotranspiration, near-surface average daily maximum and mean temperature, and precipitation data from 1979 to 2017 were derived from the University of East Anglia Climatic Research Unit (CRU) Time-Series (TS) version 4.04 dataset (https://catalogue.ceda.ac.uk/uuid/89e1e34ec3554dc98594a5732622bce9) with a 0.5° resolution [56]. To be consistent with the resolution of the GPP dataset, the nearest neighbor method was used to resample the climate data from a resolution of 0.5° to a resolution of 0.05°. CRU TS data have been widely used to explore the possible impacts of climate change in different fields of study. Many studies have also verified the accuracy of the CRU TS dataset, including studies on potential evapotranspiration [57], precipitation [57–59] and temperature [58], and proved the efficiency of the dataset in climate change studies and extreme climate event analyses. Second, GPP data were obtained from the National Science and Technology Basic Conditions Platform-National Earth System Science Data Sharing Service Platform (http://www.geodata.cn), which provides GPP data with a resolution of 0.05° from 1982 to 2017 on a time scale of 8 d. This data set was estimated by the EC-LUEEC-LUE (Eddy Covariance Light Use Efficiency) model developed by Yuan et al. [60], and the results of a verification of the dataset showed that its simulation capability exceeded that of the MODIS (Moderate Resolution Imaging Spectroradiometer) GPP product. MODIS GPP was produced by National Aeronautics and Space Administration (NASA), Washington, DC, USA. Third, China's vegetation zoning vector boundary data were obtained from the 1:1 million vegetation coverage map of China from the Data Sharing Center of the Institute of Geographic Sciences and Natural Resources Research, Chinese Academy of Sciences, Beijing, China (http://www.resdc.cn/). The zoning data reflect the detailed regional distribution and zonal differentiation of vegetation in 36 subareas of the 8 vegetation regions of China.

## 3. Method

Figure 2 is a technical flowchart of this study that mainly includes four steps: the calculations of the drought index, annual average GPP, annual GPP anomaly, yearly average of the near-surface daily mean temperature (hereinafter referred to as T), and the maximum of the monthly average daily maximum temperature of each year (hereinafter referred to as Tmax); trend analyses of annual average GPP, T, and the drought index; the construction of joint probability distribution models for GPP anomalies, Tmax, and the drought index; and the probability estimation of extreme negative anomalies of GPP (ENAG) under different drought and heat levels.

### 3.1. GPP and Extreme Negative Anomalies

An anomaly reflects fluctuations in the sample data, and positive and negative anomalies represent increases or decreases in the sample, respectively, in response to the long-term general condition of the data. According to the method proposed by Papagiannopoulou et al. [61], the removal of the linear trend of the annual average GPP time series (Equations (1) and (2)) was performed pixel by pixel as follows:

$$y_t \approx y_t{}^{Tr} = \alpha_0 + \alpha_1 t \tag{1}$$

$$y_t{}^D = y_t - y_t{}^{Tr}, \tag{2}$$

where $y_t$ is the original time series of GPP, $y_t{}^{Tr}$ is the trend series during the study period, $y_t{}^D$ is the series data without a linear trend, $t$ is the time series data during the study period, and $\alpha_0$ and $\alpha_1$ are the intercept and slope of the linear fit between GPP and the time series, respectively.

After de-trending, we got the de-trended average annual GPP time series (that is the GPP anomaly time series). Then, according to the method developed by Wang et al. [14],

ENAG was defined as GPP anomaly at least −1.5 times standard deviation ($\sigma$) lower than the mean value based on the GPP anomaly time series.

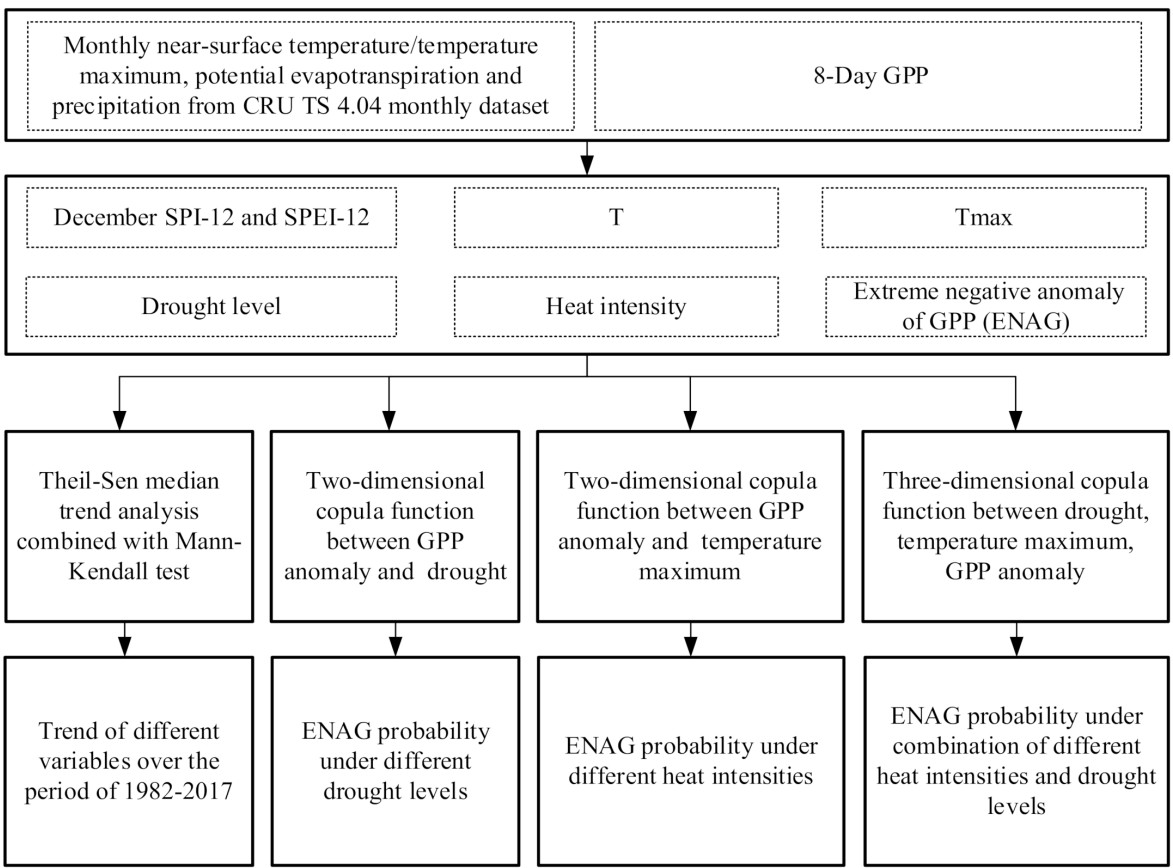

**Figure 2.** Technical flowchart of the study. CRU TS: University of East Anglia Climatic Research Unit (CRU) Time-Series (TS); GPP: gross primary production; SPI: standardized precipitation index; SPEI: standardized precipitation evapotranspiration index; T: yearly average of the near-surface daily mean temperature; Tmax: the maximum of the monthly average daily maximum temperature of each year.

### 3.2. Heat Intensities

Based on previous studies [2,15], a threshold method was utilized to define heat events. The temperature thresholds were defined as the 99th, 95th, and 90th percentiles of all Tmax values, which corresponded to severe, moderate, and mild heat, respectively. For a given year, Tmax is maximum of monthly near-surface average daily maximum.

### 3.3. Drought Indices and Levels

A large number of indices have been developed to quantify and analyze drought [62–64]. The SPI [65] and SPEI [66] are two commonly used drought indices with multiscale characteristics that are suitable for different periods and regions [67,68]. The SPI value refers to the number of standard deviations by which the observed precipitation anomaly deviates from the long-term mean precipitation [65]. The raw precipitation data are firstly fitted to a gamma or a Pearson Type III distribution, and then they are transformed to a normal distribution. Therefore, the mean SPI for the location and desired period is zero. Positive SPI values indicate rainfall surplus, whereas negative SPI values represent rainfall deficit. The SPI is based on precipitation alone and does not deal with the evapotranspiration. SPEI is an extension of the SPI and takes both precipitation and potential evapotranspiration into account when determining drought [69,70]. Thus, SPEI can capture the impact of increased temperatures on water demand. In this study, we used both SPI and SPEI. To match the

time scales of GPP and temperature series, 12-month time scale SPI and SPEI (hereinafter referred to as the SPI and SPEI) were calculated. A 12-month scale of the SPI/SPEI indicates the meteorological dryness (or wetness) of the previous 12 months as compared to historical observations. The potential evapotranspiration in the SPEI was calculated according to the FAO (Food and Agriculture Organization of the United Nations)-56 Penman–Monteith method [71]. According to the SPI/SPEI drought classification standard, the value ranges of mild, moderate, severe, and extreme drought are $(-1, -0.5]$, $(-1.5, -1]$, $(-2, -1.5]$, and $(-\infty, -2]$, respectively.

### 3.4. Trend Analysis

A combination of the Theil–Sen median trend analysis and the Mann–Kendall test was used to analyze the annual average GPP, drought index, and T trends. The Theil–Sen median can be used to calculate the median slope value of a time series, which can reduce the influence of noise on the slope and has minimal discrete data and measurement errors. The Mann–Kendall test can calculate the significance of time series changes that are not affected by outliers and is suitable for the trend testing and analysis of long time series. The Mann–Kendall test is a nonparametric statistical method that is recommended by the World Meteorological Organization and has been widely used [72]. One thing to be noted here is that hydrological and ecological time series are always affected by lag-1 serial correlation, which may overestimate the probability of detecting a significant trend [73]. Therefore, the lag-1 serial correlation was removed before using Mann–Kendall test.

### 3.5. Copula Function

Copula is a very convenient tool for constructing joint distribution and nonlinear correlation analyses with known marginal distributions [42]. It is also called the connection function. Any multivariate joint distribution can be written in terms of univariate marginal distribution functions and a copula that describes the dependence structure among variables. Suppose $x_1$, $x_2$, $\cdots$, $x_n$ are $N$ random variables, their marginal distributions are $F_{x_1}(x_1)$, $F_{x_2}(x_2)$, $\cdots$, $F_{x_n}(x_n)$, and their joint distributions are $H(x_1, x_2, \cdots, x_n)$; then there is a function $C(u_1, u_2, \cdots, u_n)$ to connect the marginal distribution and the joint distribution, such that $H(x_1, x_2, \cdots, x_n) = C[F_{x_1}(x_1), F_{x_2}(x_2), \cdots, F_{x_n}(x_n)]$. In this study, the overall joint distribution was obtained from the marginal distribution of each element. When calculating the marginal distribution of variables, it is often difficult to assume the distribution of a sequence per pixel; therefore, the kernel density estimation method [24] was used in this study to calculate the marginal distributions of the annual GPP anomaly, Tmax, and drought index for each pixel.

Copula functions are categorized into many types according to their construction methods and specific uses. The bivariate normal copula function and the bivariate t-copula function have wide ranges of applications for correlation analysis. Single-parameter Archimedes-type copula functions have simple structures and are easy to solve. Commonly used single-parameter Archimedes-type copula functions include the Frank, Clayton, and Gumbel copula functions, among which the Clayton and Gumbel copula functions can only be used to calculate the joint distribution between positively correlated variables, while the Frank copula function is applicable to both positive and negative correlations. In this study, bivariate normal, t-, and Frank copula functions of the Archimedes type were used to construct 2D joint distributions between the annual GPP anomaly and drought index and between the annual GPP anomaly and Tmax. The Euclidean distance ($D^2$) between the empirical and theoretical copula, Akaike and Bayesian information criteria was used to evaluate the accuracy of the model's fit and to select the optimal model. Smaller $D^2$ values and Akaike and Bayesian information criterion values indicate better model fits than larger values. The three-element normal and t-copula functions were used to construct the 3D joint distribution between the annual GPP anomaly, drought indices, and Tmax, and the optimal model was selected according to the $D^2$ value.

## 4. Results and Analysis

### 4.1. Trend and Trend Persistence Analysis of GPP, Temperature, and Drought Indices

According to the Theil–Sen median trend analysis and the Mann–Kendall test, the change trends of the annual average GPP, T, and drought indices from 1982 to 2017 were calculated pixel by pixel ($p < 0.05$). The results are shown in Figure 3.

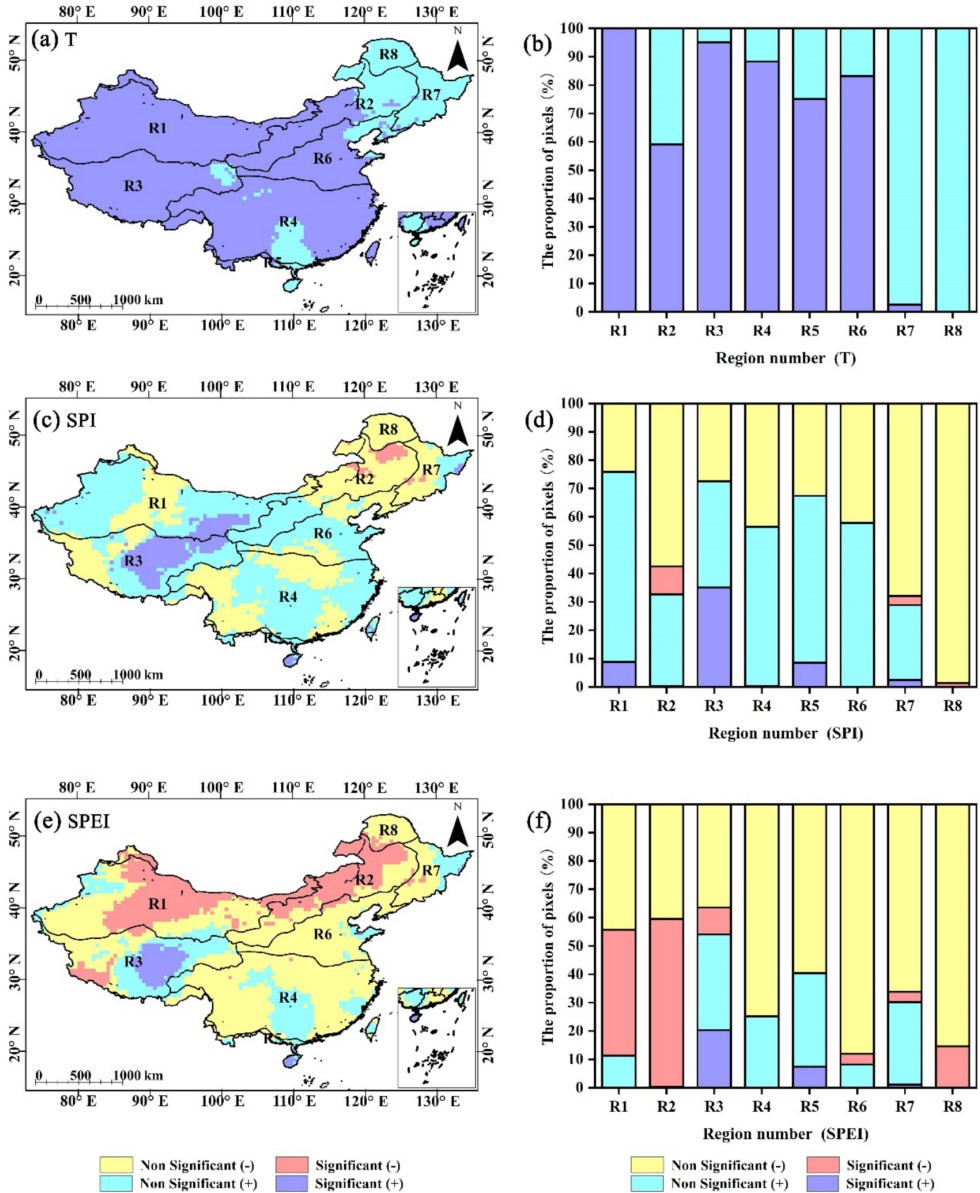

**Figure 3.** Temperature and standardized precipitation index (SPI)/standardized precipitation evapotranspiration index (SPEI) trends from 1982 to 2017. (**a**,**c**,**e**) represent the spatial distributions of the temperature, SPI, and SPEI trends, respectively; (**b**,**d**,**f**) represent the proportion of pixels with different temperature, SPI, and SPEI trends, respectively, to the total number of pixels in the eight vegetation regions of China.

In terms of the effect of global warming on temperature, except for in the northernmost parts of northeast China and southern parts of the subtropical evergreen broad-leaved forest region (R4), the temperature in all other vegetation regions showed an overall warming trend over the past 36 years (Figure 3a,b). In terms of drought, the SPI (Figure 3c) and SPEI (Figure 3e) indicated that 41.4% and 77.7% of the total pixels, respectively, contained drying trends. The pixels with significant wetting trends, derived from both the SPI and SPEI, were mainly distributed in the alpine vegetation region on the Qinghai–Tibet Plateau (R3);

however, compared with the results of the SPEI, the SPI detected more wetting pixels in the alpine vegetation region on the Qinghai–Tibet Plateau (R3) and the temperate desert region (R1) (Figure 3d,f). Additionally, compared with the results of the SPI, the SPEI detected more drying pixels, particularly in the arid and semiarid areas of central and western China (Figure 3c,e).

### 4.2. Impact of Drought on GPP

A 2D joint distribution of the annual GPP anomaly and drought index (SPI/SPEI) sequences was constructed utilizing the 2D copula function pixel by pixel, and then the ENAG probability was calculated under different drought levels for each pixel.

The ENAG probability in each vegetation region decreased with decreasing SPI drought levels (Figure 4a,c,e,g). Under extreme drought conditions, the ENAG probability reached 87.1%, and pixels with a probability greater than 60% accounted for 7.3% of the entire study area, mainly located in the temperate grassland region (R2), the subtropical evergreen broad-leaved forest region (R4), and the warm temperate deciduous broad-leaved forest region (R6). Under severe drought conditions, the areas with high ENAG probabilities were significantly reduced and mainly occurred in small portions of the temperate grassland region (R2) and the warm temperate deciduous broad-leaved forest region (R6). When a moderate drought occurred, all pixels in the study area had an ENAG probability below 40%, and when a mild drought occurred, the ENAG probability was less than 20% in all regions.

The spatial distributions of pixels representing the ENAG probabilities calculated from the SPEI (Figure 4b,d,f,h) were similar to those calculated from the SPI. Under SPEI-derived extreme drought conditions, the number of pixels with an ENAG probability occurrence greater than 80% was reduced. Pixels with high ENAG occurrence probabilities (>60%) only occurred in a few portions of the temperate grassland region (R2), the warm temperate deciduous broad-leaved forest region (R6), and the temperate coniferous and deciduous forest mixed forest region (R7); the highest occurrence probability was 83.4%. Under severe drought conditions, the highest ENAG probability was 65.2%. There was no obvious difference between the SPI and SPEI results when moderate or mild drought events occurred, but the ENAG probabilities calculated in terms of the SPI were generally higher than those calculated in terms of the SPEI.

As the drought level increased, the drought impact gradually increased in each vegetation region, and the higher the drought level was, the more evidently the ENAG probabilities increased (Figure 5). According to the total average probabilities of ENAG under different drought levels, the impacts of SPI-derived drought on GPP in different vegetation regions were determined in order from highest to lowest as follows: the temperate grassland region (R2, 68.22%) > the warm temperate deciduous broad-leaved forest region (R6, 65.56%) > the subtropical evergreen broad-leaved forest region (R4, 49.04%) > the tropical monsoon forest and rainforest region (R5, 48.13%) > the temperate coniferous and deciduous forest mixed forest region (R7, 44.87%) > the cold temperate coniferous forest region (R8, 43.20%) > the alpine vegetation region on the Qinghai–Tibet Plateau (R3, 24.99%) > the temperate desert region (R1, 23.47%). The impacts of SPEI-derived drought were determined in order from highest to lowest as follows: the warm temperate deciduous broad-leaved forest region (R6, 59.68%) > the temperate grassland region (R2, 51.71%) > the subtropical evergreen broad-leaved forest region (R4, 47.27%) > the tropical monsoon forest and rainforest region (R5, 46.00%) > the temperate coniferous and deciduous forest mixed forest region (R7, 43.03%) > the cold temperate coniferous forest region (R8, 42.28%) > the alpine vegetation region on the Qinghai–Tibet Plateau (R3, 21.16%) > the temperate desert region (R1, 20.78%).

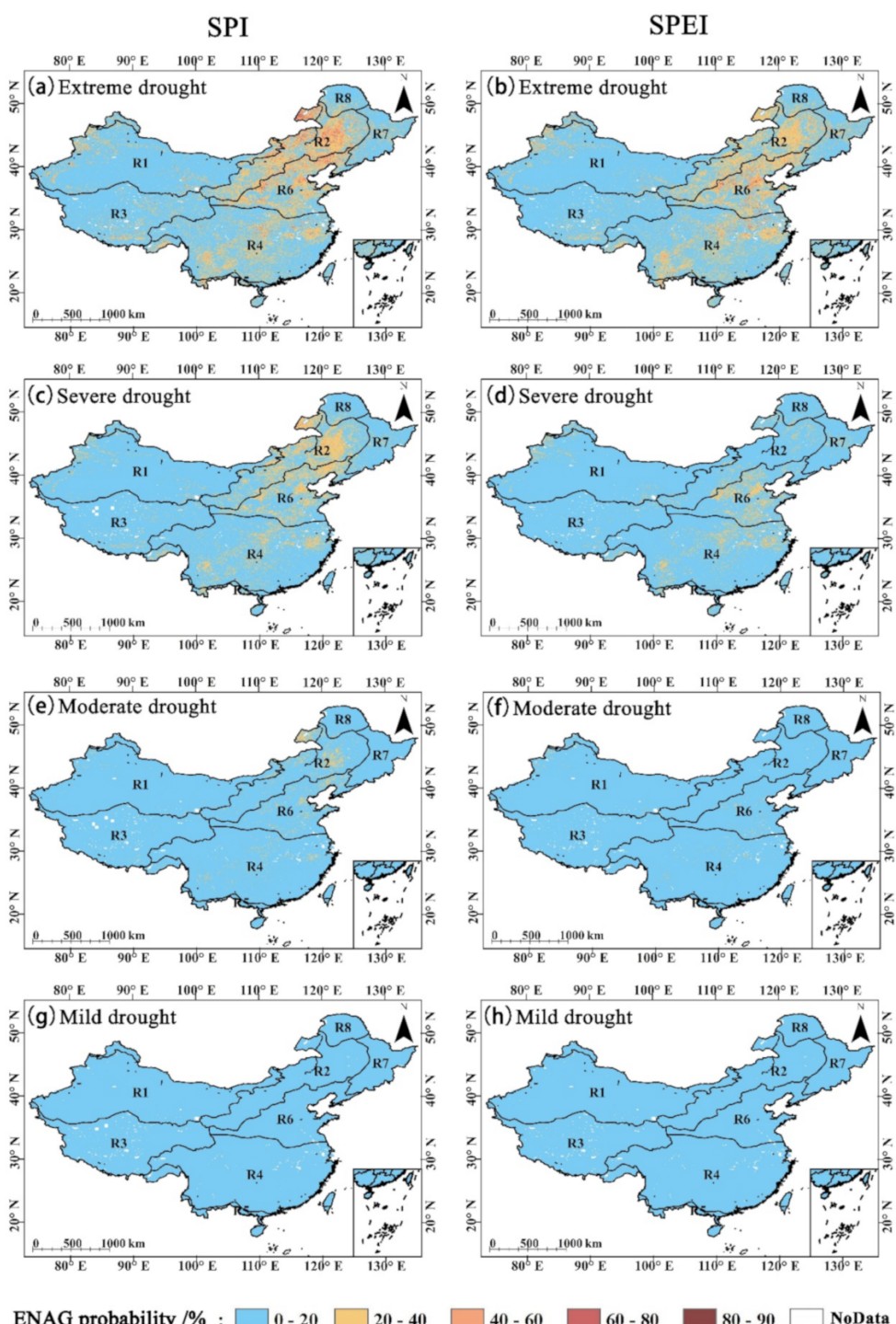

**Figure 4.** Spatial distribution of the probabilities of ENAG under different drought levels. (**a,c,e,g**) represent extreme, severe, moderate, and mild droughts classified in terms of the SPI, respectively; (**b,d,f,h**) represent extreme, severe, moderate, and mild drought classified in terms of the SPEI, respectively. (Note: The copula function cannot be built in the pixels where the GPP data are missing or the GPP value is constant in all years. Those pixels were removed and are shown as white).

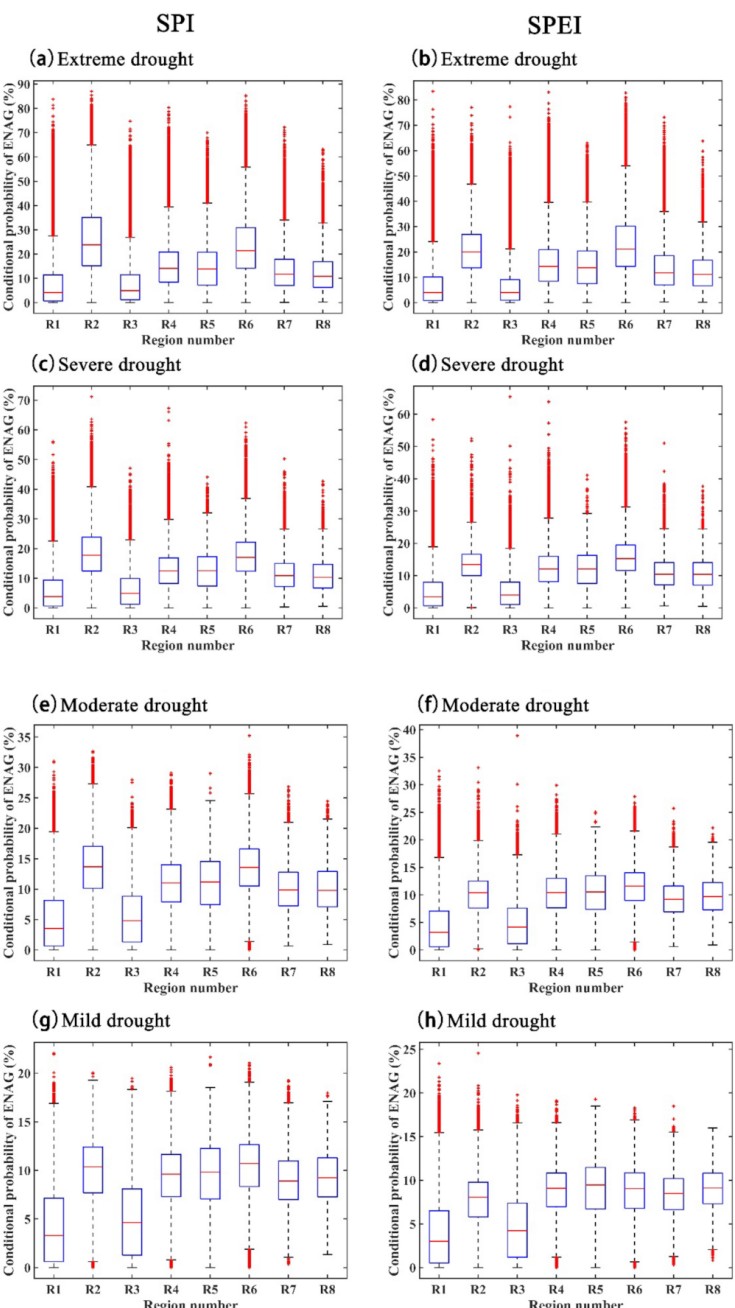

**Figure 5.** Boxplots of ENAG probabilities in each vegetation region under different drought levels. (**a**,**c**,**e**,**g**) represent extreme, severe, moderate, and mild drought classified in terms of the SPI, respectively; (**b**,**d**,**f**,**h**) represent extreme, severe, moderate, and mild drought classified in terms of the SPEI, respectively.

### 4.3. Impact of Heat on GPP

A 2D joint distribution of the annual GPP anomaly and Tmax sequences was constructed utilizing the 2D copula function pixel by pixel, and the ENAG probability (dramatic decrease in GPP) under different heat intensities was calculated for each pixel. The results are shown in Figure 6.

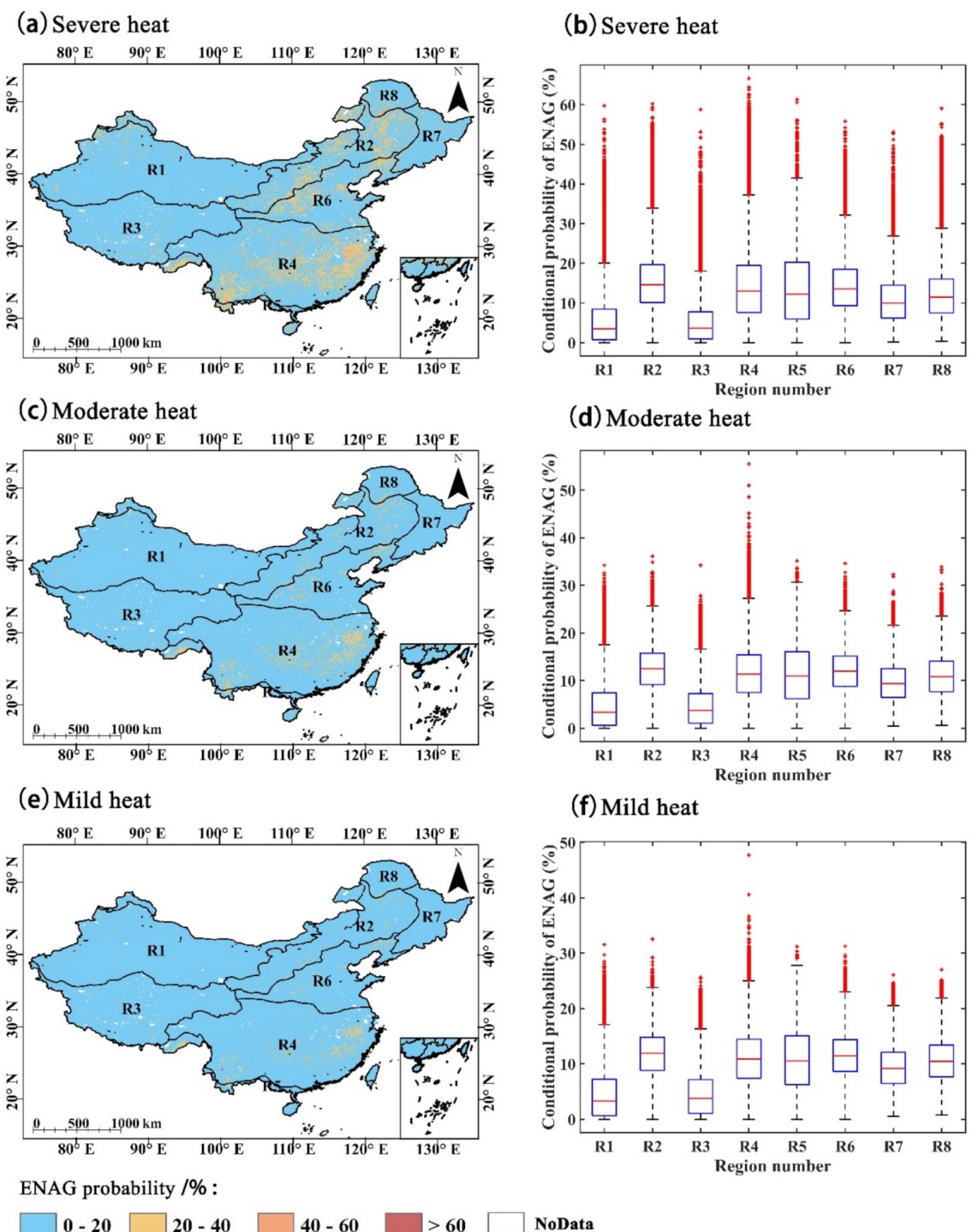

**Figure 6.** ENAG probabilities under different heat intensities. (**a,c,e**) represent the spatial distributions of the probabilities under severe, moderate, and mild heat, respectively; (**b,d,f**) represent the boxplots of ENAG probabilities in each vegetation region under severe, moderate, and mild heat, respectively (Note: the pixels displayed in white have the same meaning as Figure 4).

Generally, the ENAG probabilities under high-temperature conditions were less than those under drought conditions. Under severe heat, the maximum ENAG probability was 66.6% (Figure 6a). Pixels with ENAG probabilities greater than 40% sporadically appeared in the temperate grassland region (R2) and the subtropical evergreen broad-leaved forest region (R4), and pixels with ENAG probabilities between 20% and 40% mainly occurred

in the temperate grassland region (R2), the subtropical evergreen broad-leaved forest region (R4), and the warm temperate deciduous broad-leaved forest region (R6). When the temperature intensity was reduced from high to moderate or mild (Figure 6c,e), ENAG probabilities greater than 40% only occurred in a few pixels, mainly located in the eastern part of the subtropical evergreen broad-leaved forest region (R4). In other areas, the ENAG probabilities were generally less than 20%.

When heat intensities increased, the ENAG probabilities in each vegetation region increased, and the higher the heat intensity was, the greater the increase in the ENAG probability was (Figure 6b,d,f). In terms of the total average ENAG probability, the impact of heat on GPP in the different vegetation regions was calculated in order from highest to lowest as follows: the temperate grassland region (R2, 39.66%) > the warm temperate deciduous broad-leaved forest region (R6, 38.20%) > the subtropical evergreen broad-leaved forest region (R4, 37.28%) > the tropical monsoon forest and rainforest region (R5, 35.88%) > the cold temperate coniferous forest region (R8, 34.56%) > the temperate coniferous and deciduous forest mixed forest region (R7, 30.28%) > the temperate desert region (R1, 14.94%) > the alpine vegetation region on the Qinghai–Tibet Plateau (R3, 14.64%).

### 4.4. Comprehensive Impact of Drought and Heat on GPP

The conditional ENAG probability of each pixel under different drought levels and heat intensities was calculated by constructing a 3D joint distribution of the annual GPP anomaly, SPI/SPEI, and Tmax. The results are shown in Figures 7 and 8.

The spatial distribution of ENAG probabilities under conditions of extreme drought and severe heat (Figures 7a and 8a) is more similar to the distribution of ENAG probabilities under extreme drought (Figure 4a,b) than to the distribution of ENAG probabilities under severe heat (Figure 6a). Overall, when heat and drought occur simultaneously, the ENAG probabilities are higher and the number of pixels with high ENAG probabilities increases.

When the drought level and heat intensity weaken, the ENAG probability in each vegetation region decreases. In the SPI scenario, the maximum ENAG probability was reduced from 90.5% (under simultaneous extreme drought and severe heat conditions) to 33.3% (under simultaneous mild drought and mild heat conditions), and in the SPEI scenario, the maximum ENAG probability is reduced from 84.3% (under simultaneous extreme drought and severe heat conditions) to 32.9% (under simultaneous mild drought and mild heat conditions). Generally, the ENAG probability decreased as the drought level decreased, and the higher the drought level was, the less the ENAG probability changed with a change in the heat intensity. The difference in the average ENAG probability range between severe and mild heat among the eight vegetation regions under SPI-derived (SPEI-derived) extreme drought conditions was 5.99% (9.22%); however, there was no discernible difference in the ENAG probability distributions. Under SPI-derived (SPEI-derived) mild drought conditions (Figures 7 and 8j–l), the difference in the average ENAG probability range between severe and mild heat of the eight vegetation regions was 16.01% (15.89%), and the ENAG probability distribution showed an apparent difference with a decrease in the heat intensity.

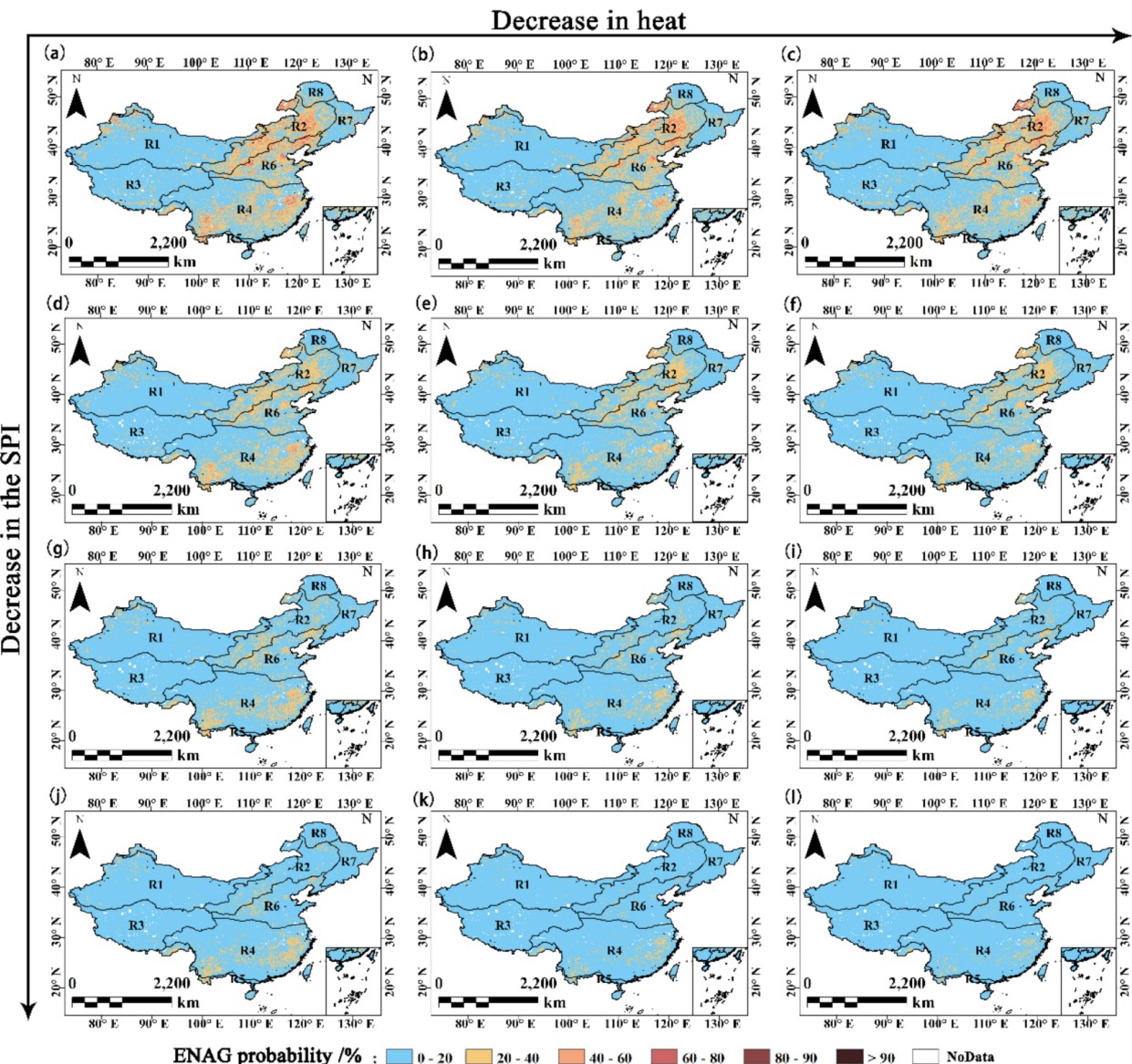

**Figure 7.** Spatial distributions of ENAG probabilities under different drought levels and different heat intensities. From left to right, the heat intensities decrease from severe to mild, and from top to bottom, the drought levels decrease from extreme to mild (Note: the pixels displayed in white have the same meaning as Figure 4).

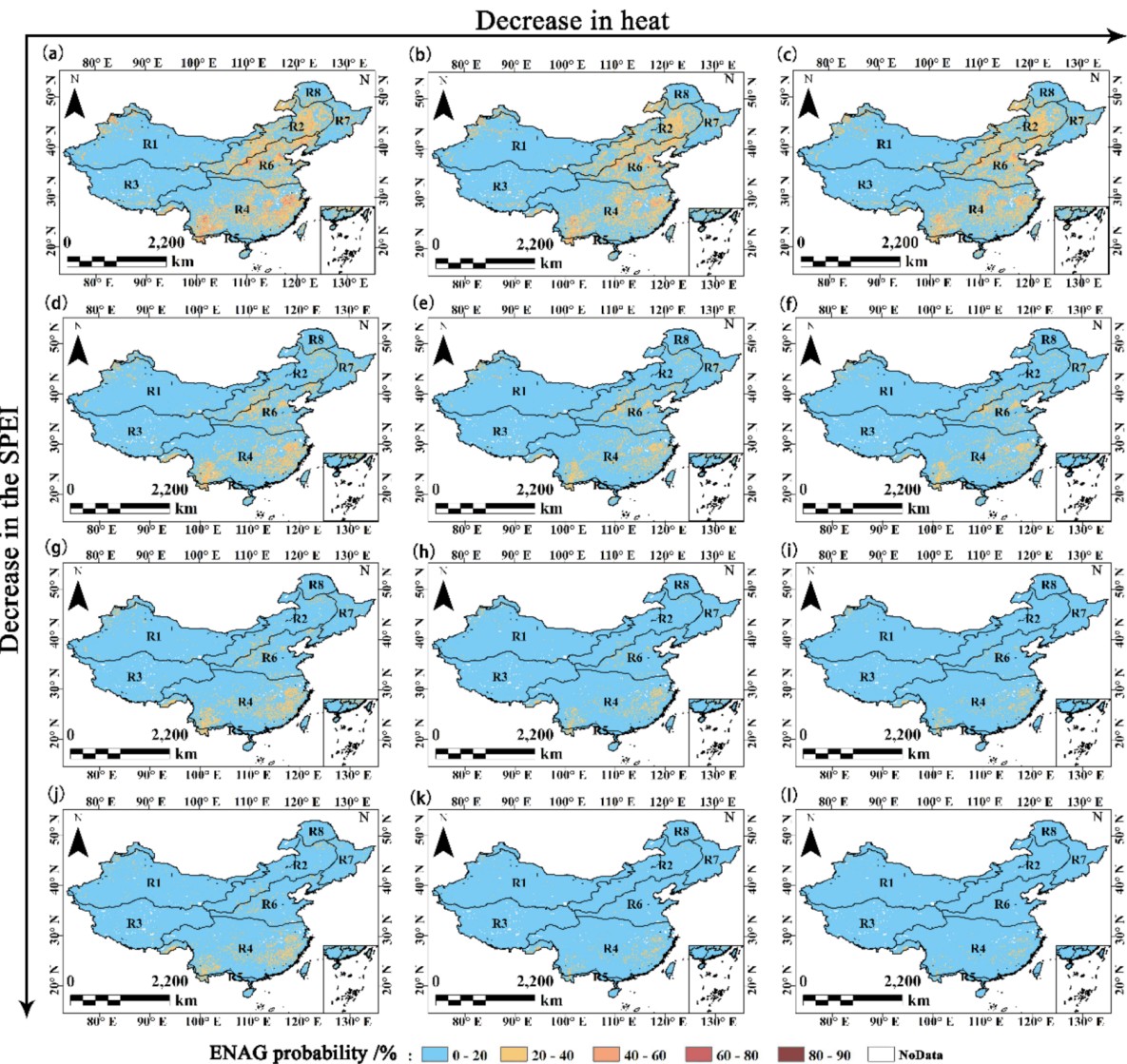

**Figure 8.** Spatial distributions of ENAG probabilities under different drought levels (SPEI) and different heat intensities. From left to right, the heat intensities decrease from severe to mild, and from top to bottom, the drought levels decrease from extreme to mild (Note: the pixels displayed in white have the same meaning as Figure 4).

## 5. Discussion

### 5.1. Comparison of the SPEI and SPI

The SPEI trend was more evident than the SPI trend. The SPEI and SPI showed that pixels with significant change trends (including wetting trends and drying trends) accounted for 24.5% and 9.6% of the total number of pixels, respectively. Compared to the SPI, the SPEI detected a drying trend in the western arid regions, particularly in the temperate desert region (R1). The main reason for this result might be that the SPEI considers the impacts of precipitation and evapotranspiration on drought simultaneously. The SPEI is very sensitive to the potential evapotranspiration calculation model and overestimates the contributions of temperature changes to drought. A previous study also indicated that the contributions of temperature anomalies to drought are often overestimated by the SPEI in arid and semiarid regions [70], whereas the SPI ignores the contributions of temperature anomalies. It is easy to cause wetness in arid and semiarid areas and dryness in humid areas. Generally, both the SPI and SPEI mainly showed drying trends in northeast China and wetting trends in the central portion of the Qinghai–Tibet Plateau. The SPI showed

stronger trends in areas with tendencies to become wet, and the SPEI indicated stronger trends in areas with tendencies to become dry.

Additionally, the difference in the average ENAG probability between the SPI and SPEI was small under mild drought conditions and gradually increased with increasing drought levels. (Figure 5). The ENAG probabilities calculated by the SPI were generally higher than those calculated by the SPEI, which indicates that GPP was affected more by changes in SPI-derived drought levels than by changes in SPEI-derived drought levels. This was possibly caused by the positive correlation between temperature and vegetation growth. All the vegetation regions in China showed significant warming trends. The positive contribution of temperature partially offsets the negative effect of drought on GPP. This reduces the ENAG probability when drought occurs so that the ENAG probabilities based on the SPEI are lower than those based on the SPI.

*5.2. Impacts of Heat and Drought on Different Vegetation Regions*

Both heat and drought can cause ENAG, but the degrees of the responses of different vegetation regions to heat and drought differ. According to the total average probability of ENAG under different drought levels (Figure 5), the temperate grassland region (R2) and the warm temperate deciduous broad-leaved forest region (R6) are the most sensitive regions to drought; in these regions, the average ENAG probability is highest under drought conditions. The main reason for this result is probably that most of the temperate grassland region (R2) is located in the semiarid zone of northern China where precipitation has a great impact on vegetation coverage. Due to the overall temperature increase, meteorological drought events have caused soil aridification. Natural water shortages have become a limiting factor restricting grassland productivity [74]. The warm temperate deciduous broad-leaved forest region (R6) is the major crop-producing region in China, and the recognized primary limiting factor for crop production in this region is drought due to low annual precipitation [75,76]. As one of important agricultural drought mitigation measures, irrigation is popular among farmers in R6 [77]. Irrigation relieves the impact of extreme climate on crops. In theory, without the mitigation effect of irrigation, the ENAG probability would be higher.

The most sensitive region to heat is also the temperate grassland region (R2), which has a simple ecosystem structure and a weak self-regulation ability. Heat intensities cause excessive transpiration and water loss in grasslands, resulting in metabolic imbalance. The subtropical evergreen broad-leaved forest region (R4) and the tropical monsoon forest and rainforest region (R5) in southern China are also sensitive to heat because these regions are located in subtropical and tropical monsoon climate zones with adequate water and heat resources. When heat occurs, the stomata of vegetation are closed to prevent excessive transpiration of water, resulting in a decrease in the photosynthetic rate, which directly causes the light reaction to be inhibited. The carbon dioxide absorbed by leaves is also reduced under heat, and the dark reaction is inhibited. Heat at night accelerates organic matter decomposition, which is not conducive to organic matter preservation. All these factors contribute to the reduction in photosynthesis efficiency under high-temperature conditions and affect vegetation growth. Chen et al. [2] studied the effects and drivers of negative extreme events on gross primary productivity and pointed out that extreme GPP events in southern China were mostly associated with temperature extremes, including extreme heat.

Both heat and drought have the smallest impacts on the alpine vegetation region on the Qinghai–Tibet Plateau (R3) and the temperate desert region (R1). Alpine vegetation has high altitudes and low temperatures all year, both of which severely limit the productivity of most alpine grasslands at all growth stages, and, appropriately, heat is conducive to vegetation growth in this area. The latest research of You et al. [49] showed that GPP was positively correlated with temperature across most parts of the grasslands of R3 during both the entire growing season and different growth stages. Precipitation that affects vegetation growth mainly occurs in warm weather periods [78,79]. Additionally,

the R3 region shows a significant wetting trend, and the drought impact in this region is relatively restricted compared with that in the other regions. The R1 region has a temperate, extremely arid climate with sparse rainfall. The typical vegetation in this region is temperate desert vegetation, which generally exhibits heat resistance and xerophytic characteristics; therefore, R1 is not sensitive to heat and drought.

In all the vegetation regions, the ENAG probability caused by heat is much smaller than that caused by drought. The impact of extremely heat on GPP is negligible without a long-term duration or drought stress (a water shortage) [15]. When heat and drought occur simultaneously, the ENAG probability increases in most vegetation regions compared to that under the impact of either heat or drought alone. The temperate grassland region (R2), the subtropical evergreen broad-leaved forest region (R4), and the warm temperate decidu-ous broad-leaved forest region (R6) have the highest ENAG probabilities (Figures 7 and 8), which may cause fragile ecological environments and continuous ENAG occurrences. In addition, the combined effects of heat and drought are not simply superimposed. When a strong drought occurs, vegetation productivity is severely reduced; therefore, heat stress will not cause further GPP loss. When the drought level is reduced to a moderate or mild drought, the heat stress impact on the GPP is enhanced (Figures 7 and 8).

*5.3. Research Contributions*

First, this study introduced copula functions into the analysis of the impacts of drought and heat on GPP. By constructing a 2D joint distribution of GPP and heat/drought and a 3D joint distribution of GPP, heat, and drought, ENAG probabilities under different drought levels and heat intensities were obtained. Generally, previous research on the impacts of extreme climate events on GPP has been divided into two methodologies. In the first method, drought or heat events that occurred in a certain period and region were selected with which to monitor dynamic GPP changes under extreme climate conditions in the given period compared with changes in other periods. In the second method, extreme events, including extreme climatic events (heat, heat waves, extreme precipitation, extreme cold, etc.) and extreme GPP events were first identified, and the relationship between extreme climate events and extreme GPP was then analyzed by correlation analysis, regression analysis, or other methods. Neither of the above methods consider interactions between climate variables. The research method utilized in this study analyzed the influence of drought and heat stress on GPP more intuitively from the perspective of probability and considered the combined influence and synergistic effect of these two influences on GPP. This methodology is conducive to understanding the possible results of the distribution of GPP losses under extreme climate conditions and to providing a reference for future research.

Second, this study reveals the nonlinear impacts of drought and heat on GPP. Accord-ing to the 2D copula model of GPP and drought/heat, this study found that when the drought level/heat intensity increases, the ENAG probability in each vegetation region increases, and the higher the drought level/heat intensity is, the more the probability increases. According to the 3D copula model, it was determined that the combined impact of drought and heat on vegetation growth is not a simple linear accumulation. When the drought level is low, heat has a big impact on GPP; however, when the drought level is high, the impact of heat on GPP is limited, which is of great significance for future research on the responses of vegetation to heat and drought.

Third, to study the interaction between vegetation and climate and to evaluate the impact of climate change on vegetation, vegetation models have been rapidly developed. A dynamic global vegetation model is a standard vegetation simulation model that mainly uses climate data and $CO_2$ concentrations as inputs to simulate the physiological processes of vegetation and to allow for the quantitative understanding of the interaction between vegetation and climate through coupling with climate models [80]. However, the climate data utilized in this vegetation model mostly consist of precipitation and temperature data that are transformed into environmentally limiting factors through functions that

eventually become model parameters. This method often ignores interactions between factors, such as between drought and heat, and results in the limiting effect of drought on heat being overlooked, thus causing additional uncertainties in the simulation results if they are considered separately. Therefore, future developments in vegetation simulation models and improvements in simulation accuracy should focus on the synergistic effects and potential nonlinear relationships between different stresses (such as drought and heat).

*5.4. Limitations and Research Prospects*

This study divided the research area into eight vegetation regions for analysis; however, the responses of vegetation to heat and drought are very different in each region due to factors such as climate conditions, topographic differences, and vegetation types. Furthermore, this study only focused on the impact of two types of extreme climate events, heat and drought, which have universal coupling effects on GPP anomalies. However, vegetation growth can be influenced by many other extreme climatic conditions, such as flooding [81], low temperatures, and cold damage [82].

Additionally, at present, there is no unified definition of heat. Academia generally believes that heat means that the temperature exceeds a certain threshold (including absolute thresholds and relative thresholds), and the absolute threshold defines heat events in terms of specific temperatures. Heat defined by absolute thresholds has more direct and obvious influence on vegetation and increases the comparability between the responses of different vegetation types to heat. However, it is difficult to obtain accurate heat threshold data for different vegetation types due to the rich vegetation types in different vegetation regions. Therefore, this study applied the 99th, 95th, and 90th percentile threshold values of temperature anomaly sequences to define the heat intensities. This method conveniently provides an adaptive heat definition that requires a long temperature time series without frequent small climatic changes. Otherwise, the use of this method may cause uncertainty in experimental results. Therefore, the influence of different definitions of heat on research results and comparisons between different definitions should be considered in future studies.

Finally, this study used copula functions to estimate ENAG probabilities under different heat intensities and drought levels; this method is more objective than traditional nonprobabilistic evaluation methods. However, copula functions are difficult to use in higher-dimensional research, such as in studies on the synergistic effects of heat, freezing, heavy rain, and drought on GPP. In addition, different vegetation types have different responses to the time of occurrence of extreme climate events, and the responses of vegetation to extreme climate events have certain hysteresis effects. This study did not identify the time of occurrence of extreme climate events, nor did it distinguish between the growing seasons and nongrowing seasons of vegetation. Considering the following three points, our study was conducted on an annual scale. First, if analyzing the GPP of the growing season, we must first determine the start and end time of the growing season of vegetation in different regions (or even each pixel). The growing season of different vegetation in the same area is different. For a given vegetation type and in the same region, its growing season also varies in different years. There are also errors in determining the growing season. Second, the responses of vegetation to extreme climate events have certain hysteresis effects. The drought before the growing season will affect the surface water storage, and then it will affect the growth and development of vegetation [83,84]. Third, a drought index with a 12-month time scale is often used in studies on the relationship between drought and vegetation response [85–88]. However, future studies that comprehensively consider these factors and conduct more detailed analyses, such as identifying various extreme climate events within a vegetation region and analyzing their impacts on vegetation productivity within a certain time window, are necessary.

## 6. Conclusions

In this study, based on GPP data from the EC-LUE model and CRU TS 4.04 climatic (temperature and precipitation) data, we analyzed the trends of temperatures and drought indices (SPI/SPEI) in China and investigated the impacts of extreme climatic conditions (drought and heat) on GPP by constructing multidimensional copula functions between GPP and climate variables. The results showed that the SPI/SPEI showed an increasing drought trend in 41.4%/77.7% of the total pixels, and the temperatures in each vegetation region increased significantly. The impact of drought on GPP is greater than that of heat, and their combined impact is greater than the impact of either single stress factor alone. When the drought level is high, drought will limit the effects of heat on vegetation. Our results implied that policymakers could pay more attention to GPP deficits in the temperate grassland region (R2) and the warm temperate deciduous broad-leaved forest region (R6) because they are most vulnerable to extreme events, including drought and heat, to mitigate the potential impacts of future climate extreme events.

**Author Contributions:** Conceptualization, X.Z.; methodology, X.Z.; resources, X.Z.; formal analysis, S.Z.; data curation, S.Z.; writing—original draft preparation, S.Z.; writing—review and editing, T.L. and Y.L.; All authors have read and agreed to the published version of the manuscript.

**Funding:** This work was supported by the National Key R&D Program of China (Grant No. 2019YFA0606900) and the National Natural Science Foundation of China (Grant No. 42077436).

**Institutional Review Board Statement:** Not applicable.

**Informed Consent Statement:** Not applicable.

**Data Availability Statement:** Data is available upon request.

**Conflicts of Interest:** The authors declare no conflict of interest.

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
