# Peer review of "Impacts of Heat and Drought on Gross Primary Productivity in China"

_remotesensing, doi:10.3390/rs13030378_

Round 1

Reviewer 1 Report

The manuscript by Zhu et al. (Manuscript ID: remotesensing-1054443) aims at answering an important question, i.e. how high temperature and drought will jointly affect vegetation productivity in China. The research provides useful implications for ecological protection and reducing the negative impacts of global warming. The methodology of the study is novel and they showed some interesting results. In general, I think this is a good study and the manuscript is concise and well organized. The content of the paper fits the aim and scope of the journal of Remote Sensing. However, I do have some comments below, which may help further improve this manuscript. I recommend a minor revision of the paper before it is published.   

  1. There have been some similar studies in the literature, which also focus on the combined effects of high temperature and drought on vegetation and plant health, e.g. Dong et al. (2019), Young et al. (2017). I suggest the authors search for more related publications and mention these previous studies, which may help justify why it is important to investigate the drought impact on vegetation under a warming climate.

Dong, C., MacDonald, G., Okin, G. S., & Gillespie, T. W. (2019). Quantifying Drought Sensitivity of Mediterranean Climate Vegetation to Recent Warming: A Case Study in Southern California. Remote Sensing, 11(24), 2902.

Young, D. J., Stevens, J. T., Earles, J. M., Moore, J., Ellis, A., Jirka, A. L., & Latimer, A. M. (2017). Long‐term climate and competition explain forest mortality patterns under extreme drought. Ecology letters, 20(1), 78-86.

  1. Line 166-168. The sentence is confusing, and please improve and clarify.
  2. Line 185-192. As I know, hydrological and ecological time series are always affected by lag-1 serial correlation (see Yue et al. 2002), which may overestimate the probability of detecting a significant trend. Thus, I would suggest removing the lag-1 serial correlation before using Mann-Kendall test. For example, the R package “zyp” (https://cran.r-project.org/web/packages/zyp/index.html) provides a solution of whitening the serial correlated time series.

Yue, S., Pilon, P., Phinney, B., & Cavadias, G. (2002). The influence of autocorrelation on the ability to detect trend in hydrological series. Hydrological processes, 16(9), 1807-1829.

  1. I would suggest changing Fig. 5 and Fig. 6b.d.f from bar plots to box plots, which may provides more information to the readers.

Reviewer 2 Report

Please see my comments in the attached document.

Round 2

Reviewer 2 Report

Thank you for addressing my comments.